



# What drives spatial variance of hydrological behaviour? An analysis of the regional groundwater-stream continuum

Gunnar Lischeid[1,2], Justus Weyers[2], Helen Scholz[3]

[1]Research Area 3, Leibniz Centre for Agricultural Landscape Research, Müncheberg, 1537, Germany
5 [2]Institute of Environmental Science and Geography, University of Potsdam, Potsdam, 14476, Germany
[3]Department 8 – Hydrological Services, Bavarian State Authority of the Environment, Hof, 95030, Germany

*Correspondence to*: Gunnar Lischeid (lischeid@zalf.de)

**Abstract.** A sound understanding of regional scale hydrological processes and influencing factors is indispensable for
10  sustainable water resources management. It requires differentiation between natural heterogeneities, direct anthropogenic
effects, and climate change impacts. In addition, an integral perspective is required comprising both surface and groundwater
bodies. This study aimed at determining the key drivers for the spatial variance of hydrological behaviour, at gaining an
understanding why long-term trends of observed behaviour often differ in spite of spatial proximity and similar boundary
conditions, and at investigating the added value of merging stream discharge and groundwater head data for the analysis.

15  A set of 292 time series of stream discharge and groundwater head from a 36,000 km2 region in South Germany, covering a
43 years period, was subjected to a principal component analysis. The first six components were analysed in more detail. All
together they explained 77.8% of the total variance. The first component grasped mean behaviour. Three components
reflected various facets of climate patterns. Land use effects were not found to be significant when the common dependence
of land use and hydrology on climate patterns was factored out. Two further components described the damping of the
hydrological input signal in the subsurface. One of these differentiated between porous substrates and fractured or karstified
hardrocks. Damping of the input signal was very closely related to direction and strength of long-term trends. Trends were
the most clearly visible in deep groundwater time series which are suggested to be used as early warning indicators with
regard to climate change rather than shallow groundwater or stream discharge. In general, the combined analysis of stream
discharge and groundwater head proved to be very efficient, benefitting from complementary sensitivities toward single
processes and effects, and is highly recommended for future analyses.

## 1 Introduction

For climate change risk assessment of water resources a sound understanding of both general as local conditions is required.
The latter is usually based on monitoring of temporal patterns of hydrological behaviour at numerous sites. Scientists and
water resources managers are then confronted with the task to relate observed heterogeneities between sites to the respective
influencing factors. In particular, direct anthropogenic effects need to be identified and to be delineated from natural



variability. In addition, anticipated climate change effects need to be tagged. In that regard proven early warning indicators would be very helpful.

The European Water Framework Directive (WFD) requires a regular inspection of the "good quantitative" status of water bodies. To that end usually trend analyses of long-term hydrological time series are performed. They often yield inconsistent
results, often strongly differing even between adjacent sites and in spite of similar boundary conditions and of a lack of evidence of any direct anthropogenic effects (e.g., Tsypin et al., 2024). In Central Europe a series of dry and warm years started in 2018, resulting in pronounced declines of groundwater head, lake water level and terrestrial water storage at large (Bakke et al., 2020; Boergens et al., 2020). Hydrological response differed widely between sites (Bakke et al., 2020). During the last years recovery was observed at many sites, although at different speed and to different degrees. In contrast, Wunsch
et al. (2022) predict a continuous decline of groundwater levels in Germany for the next decades. In fact, gravimetric data of the GRACE mission indicate an earlier onset and a still ongoing decrease of total water storage (Xanke and Liesch, 2022). This discrepancy is all the more critical as global hydrological models failed to reproduce long-term changes detected by GRACE in various regions of the globe (Scanlon et al., 2018), and models differ substantially in terms of groundwater recharge prediction (Gnann et al., 2023). This raises serious concerns about our ability to predict and to determine climate
change impacts on hydrological processes and urgently calls for clarification.

Local assessment of water resources by authorities as well as by scientists should rely on a sound scientific basis. The last decades have witnessed a plethora of studies aiming at a better understanding of the drivers of spatial heterogeneity of stream flow behaviour (Tarasova et al., 2024). Current knowledge comprises a variety of factors of influence, derived from numerous studies, usually focussing on single or a few effects. For real world settings differentiating between and weighing various
effects is required. To that end, a major source of information is the observed variability of behaviour at different sites by relating the observed behaviour in terms of the temporal evolution of stream discharge or groundwater head to catchment characteristics.

Respective studies usually agree in terms of the role of climate indicators, that is, mean values of precipitation and evapotranspiration, their spatial and seasonal patterns, the percentage of snow, the duration and timing of snow cover, etc.
(Tarasova et al., 2024; He et al., 2024). Beyond that, however, results are far from unambiguous. Often land use/land cover or vegetation patterns have been identified as key indicators at various scales (Zhao et al., 2010; Baroni et al., 2013; He et al., 2024). Topography-related predictors proved valuable predictors in some studies (Zhao et al., 2010; Tarasova et al., 2024; Liu et al., 2024). The prominent role of soil properties has been emphasized, e.g., by Vereecken et al. (2022), Zhao et al. (2010) and Joshi and Mohanty (2010), which is vehemently disputed by Gao et al. (2023). Rather few studies tried to
relate geology and aquifer properties to stream flow patterns (Dal Molin et al., 2020; Liu et al., 2024). Although many studies reported on successful applications of various indicators in terms of better understanding and prediction of spatial patterns of stream flow dynamics, others raise serious doubts on their general usefulness beyond single case studies (Jakisch et al., 2021; Gao et al., 2023; Istalkar and Biswal, 2024).



Streams and groundwater are different facets of the same hydrological system. Nevertheless, there is a clear dichotomy in terms of scientific communities and authorities, scientific journals, conferences, and models (Berkowitz and Zehe, 2020). This seems in the first place to be due to historical reasons and path dependencies rather than on hard scientific reasons. In fact numerous studies show that explicit consideration of the tight coupling between streams and groundwater head is a prerequisite for a sound system understanding (e.g., Berghuijs and Slater, 2023). Here we go a step further and hypothesize, that stream discharge and groundwater head dynamics are nothing than two poles along a common gradient, being subject to the same processes, although at different degrees. Consequently we hypothesize that much can be learnt about groundwater processes from stream discharge dynamics, and vice versa.

In particular, the study aims

- at determining the key drivers for differing dynamics of discharge and groundwater head at different sites;
- at gaining an understanding why long-term trends of discharge and groundwater often differ in spite of spatial proximity and similar boundary conditions,
- at assessing in a systematic way similarities and dissimilarities between stream discharge and groundwater head dynamics.

Empirical studies on hydrological behaviour face the challenge how to determine hydrological behaviour in a quantitative way. It has been suggested to use selected features of the hydrographs as "signatures" which should be "meaningful" (Gupta et al., 2008). They need to be defined in advance but are hardly ever checked for relevance. Here we use an approach that makes full use of the total information content provided by hydrological time series without any arbitrary pre-selection. To that end we apply principal component analysis to a merged data set of stream discharge and groundwater head time series from a region that exhibits high heterogeneity in terms of topography, land use, and geology.

## 2 Data

The study region comprises the northern half of the Federal state of Bavaria in South Germany, between 48.5° and 50.5° north latitude and 9.0° and 13.6° east longitude (Fig. 1) with a total area of 36,000 km$^2$. To rule out effects of melting glaciers and distinct nival zones, the region south of the Danube River with tributaries from the Alpine region was excluded from the analysis. Altitude of the selected region varies between 100 m a.s.l. in the Northwest and 1456 m a.s.l. in the Southeast (Fig. 1). Geological strata in the subsurface encompass a wide range of various geological units from Precambrian hardrocks to Quarternary deposits. Correspondingly, both porous and fractured or karst bedrock aquifers are found.



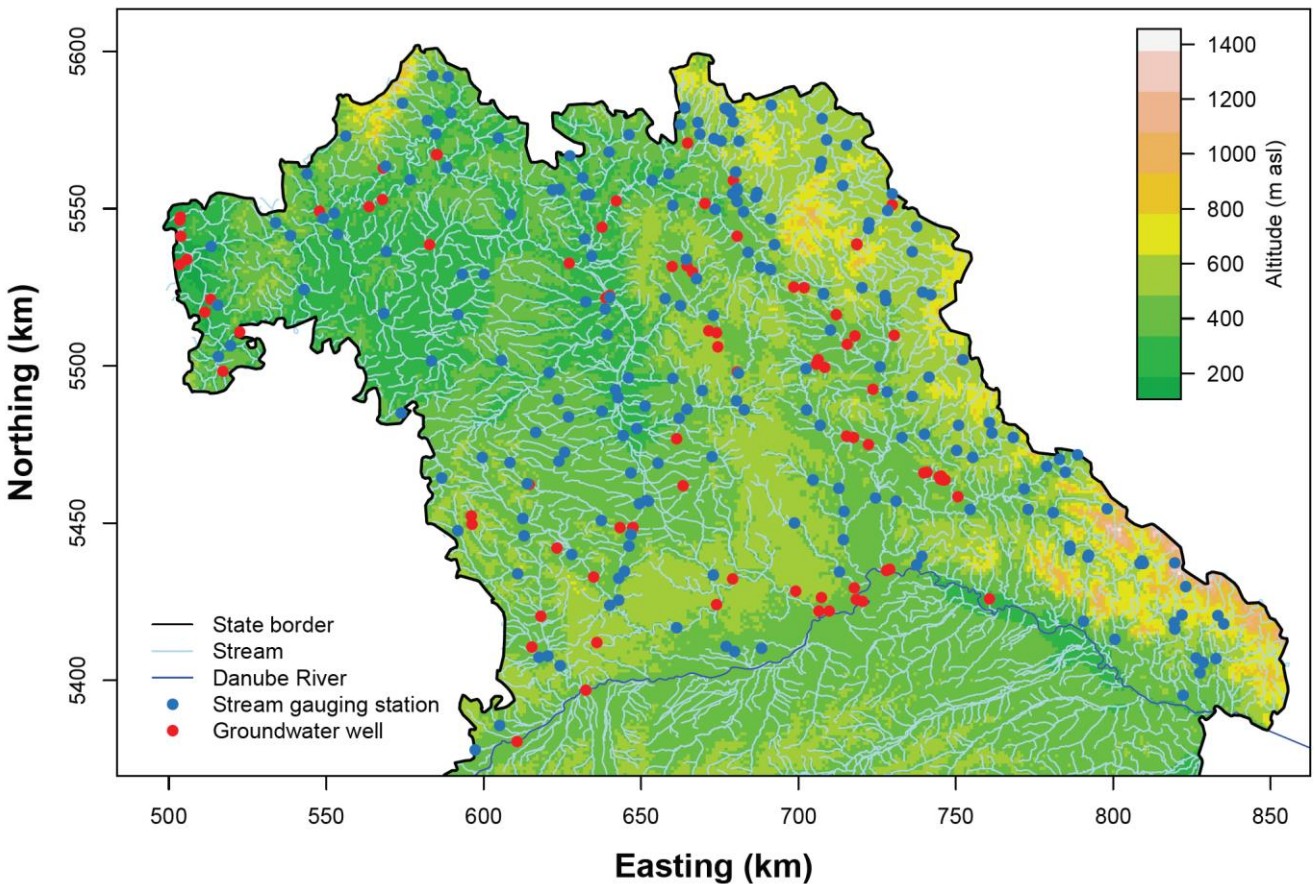

Figure 1: Location of the sampling sites in the Northern half of the Federal State of Bavaria, Germany.

Annual mean temperature varied between 4.3°C (Großer Falkenstein mountain) and 9.9°C (town of Kitzingen), and annual mean precipitation between 601 mm/a (town of Würzburg) and 1510 mm/a (Großer Arber mountain) from 1981 to 2010 (Deutscher Wetterdienst 2024). The region is characterized by a heterogeneous mosaic of land use classes, where arable fields prevail in the lowlands, and forests at higher altitude. Mean population density is about 160 population per km$^2$, which is about 67% that of Germany and 146% that of the European Union. The largest town of the region is Nürnberg with

526,000 inhabitants.

The study was performed on a set of time series of stream discharge and groundwater head of the long-term monitoring program run by the Bavarian State Authority of the Environment (LfU Bayern). Measurement sites were selected to rule out direct strong anthropogenic impacts. The data covered a period of 43 full calendar years 1980 – 2022. For the principal component analysis a large data set of synchronous and gapless time series was required. Measurements were aggregated to

daily mean intervals. Thus time series with coarser time resolution or with data gaps needed to be interpolated. To minimise any resulting bias an autocorrelation analysis was performed for each time series separately. To account for irregular



sampling intervals or data gaps, autocorrelation was determined for blocks of seven (discharge) or ten days (groundwater head) lag width, analogously to the approach used for semivariograms in geostatistics. In a next step a series of target dates was defined, that is, every week's Monday from January 1, 1980, to December 31, 2022, summing up to 2243 days of
measurements per site. For each site a corresponding time series was established by linear interpolation to the target dates as long as distance to nearest day in the observed data did not exceed the lag width of autocorrelation of 0.5. Thus the risk was minimised that interpolation would smooth out existent structures like interim peaks. Consequently, for smooth time series longer data gaps were accepted as for very responsive time series. Eventually, gapless time series were generated for 207 gauging stations and 85 groundwater wells, summing up to 292 time series in total.

Shapefiles of the river catchments were provided by the Bavarian State Authority of the Environment (LfU Bayern). Catchment area varied between 3 km2 and 23,031 km$^2$ with a median of 164 km$^2$, and the first and third quartile of 74 km$^2$ and 412 km$^2$. Stream network data were downloaded from OpenStreetMap (2023). Digital elevation data at 200 m and 1000 m resolution were provided by Bundesamt für Kartographie und Geodäsie (2023; 2024). Land use was accessed using Corine data as of the year of 2000, that is, from the middle of the study period 1980 – 2022, compiled by Umweltbundesamt
(2016). Raster data of mean climate data for the 1980-2022 period at 1 km resolution were downloaded from Deutscher Wetterdienst (2023), including precipitation, number of snow days (snow cover of 1 cm at least measured in the morning), and potential evapotranspiration according to Allen et al. (1998).

Based on these data the catchments of the gauging stations were characterized by mean elevation, 1980-2022 mean values of meteorological variables, and the share of arable land, grassland, forest, wetlands, freshwater and built-up area.
Correspondingly, land use was determined within a radius of 500 m around each groundwater well.

**3 Method**

Principal component analysis (PCA) of time series has been widely used in atmospheric sciences since the late 1940s where it is termed the "Empirical Orthogonal Functions" approach (Hannachi et al., 2007). It is also known as "Karhunen-Loève decomposition" in statistics (Joliffe, 2002). It is now increasingly used for analysis of multi-temporal remote sensing data. In
contrast, it has rarely been used in hydrology. Gottschalk (1985) presented an early application to stream flow data, and Longuevergne et al. (2007) to groundwater head data.

Application of PCA to data sets of time series can serve various purposes. E.g., in atmospheric sciences and remote sensing it is often used to determine the "key features" in comprehensive data sets. In this study PCA is applied to identify and delineate single factors of influence in a set of time series that are subjected to numerous effects synchronously, and to relate
these components to processes. In mathematical terms PCA performs an eigenvalue decomposition of the covariance matrix of the set of time series into a set of orthogonal components that are used to generate orthogonal (i.e., uncorrelated) synthetic time series of the same length as that of the genuine observations. Each of these components is assigned an eigenvalue which



is proportional to the share of variance of the total data set explained by the respective component. Principal components are sorted by eigenvalues in descending order.

For PCA the time series of observations need to be synchronous, that is, to have the same time axis. To ensure equal weighing of all time series irrespective of absolute values and amplitudes, observed data are normalized to zero mean and unit variance prior PCA. Thus, information about absolute values is not considered by PCA.

Assigning principal components to real-world processes is partly based on the analysis of loadings, that is, bivariate correlation between components and time series. Note that in this study loadings were not weighted by eigenvalues of the

respective principal component, unlike in some other studies. In addition, the time series of scores of the component are used for identification of the respective processes. When PCA is applied to a regional set of measurements the same or similar observables, the first principal component (PC1) often grasps the mean behaviour, that is, averaged over all sites. Subsequent principal components then describe typical patterns of deviation from mean behaviour at single sites due to specific effects. Thus to ease interpretation, plots of two synthetic time series are studied for each principal component, one generated by

adding the effect of the respective principal component to that of PC1, and the other by subtracting it from PC1. Thus the effect of the respective principal components can be studied, e.g., with respect to shifting seasonal patterns, reducing or increasing peak values, etc. compared to mean behaviour.

All analyses have been performed and all graphs have been generated in R (R Core Team, 2022) using different versions (4.1.3, 4.3.0, 4.3.1) and the Kendall (McLeod, 2011), proj4 (Urbanek, 2022), sf (Pebesma, 2018), raster (Hijmans, 2023),

and vioplot (Adler et al., 2022) package.

## 4 Results

Here only the first six principal components are described in more detail which account for at least 2% of explained variance each. Altogether they explain 77.8% of the total variance. The distributions of loadings for stream discharge and groundwater head data taken together are approximately symmetrical with respect to the mean value for most principal

components (Fig. 2). However, the first principal component (PC1) is a major exception where only one single groundwater head time series exhibited a slightly negative loading. Significant differences between loadings of stream discharge and groundwater head (Wilcoxon text, $p < 0.05$) were found only in seven out of 292 principal components in total (PC1, PC2, PC4, PC6, PC9, PC12, PC14). Based on the eigenvalues of these principal components, PC1 and PC2 together explain 87% of the differences between these two groups. Beyond that, the two groups differ in terms of the variance range of loadings:

PC3, PC5 and PC6 exhibit larger variance for the discharge time series, and PC2 and PC4 for groundwater head time series.




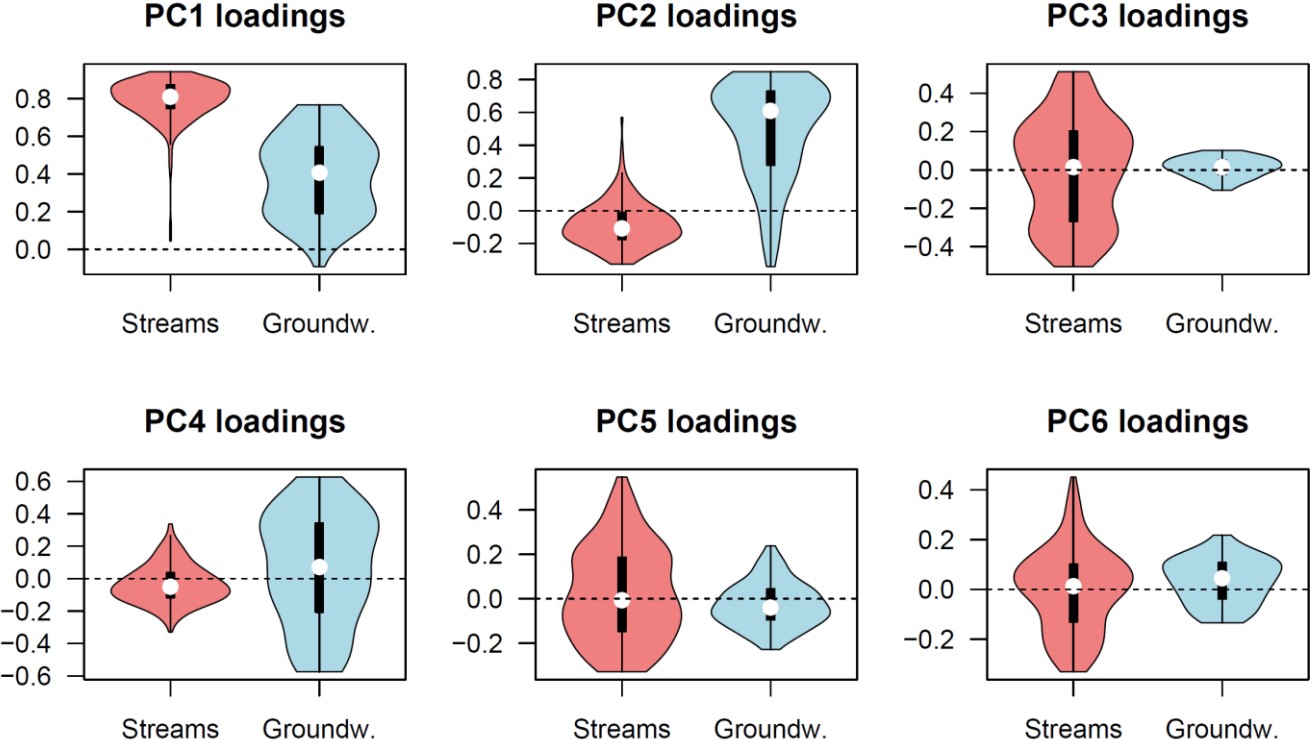

**Figure 2: Loadings of the studied time series on the first six principal components, differentiating between stream discharge and groundwater head.**


### 4.1 PC1: Mean behaviour

The first principal components grasped 51.1% of the total variance of the data set (Table 1). Scores of PC1 are very closely related to the time series of means of all z-normalized observed time series (r = 0.96), as has been often found in similar studies (Hohenbrink et al., 2016; Lischeid et al., 2021; Thomas et al., 2012). Note that differing from the usual practice of EOF application to anomalies in climatology, mean behaviour had not been subtracted beforehand in this study. Whereas PC1 grasps the communalities of most of the observed time series, subsequent principal components describe typical patterns of deviation from the time series of PC1 scores, e.g., in terms of amplitude size, shift in seasonal patterns, or the like.

Time series of stream discharge mostly exhibit very high positive correlation with this mean behaviour (Fig. 2). But that holds only for a minor proportion of groundwater head time series. Part of the bias of PC1 toward stream discharge is due to the fact that these time series constitute 71% of the total number of analysed time series. On the other hand, though, groundwater data exhibited much more pronounced variance of PC1 loadings. Thus, differences between time series of ground water head are much more pronounced compared to those of stream discharge.



**Table 1: Overview of the first six principal components**

| Principal component | Explained variance | Cumulative explained variance | Interpretation |
|---|---:|---:|---|
| PC1 | 51.1% | 51.1% | Mean time course |
| PC2 | 11.3% | 62.5% | Damping of hydrological input signals |
| PC3 | 5.3% | 67.7% | West-east climatic gradient |
| PC4 | 4.4% | 72.2% | Loose sediments vs. karstified or fractured bedrock |
| PC5 | 3.4% | 75.6% | Snow pack |
| PC6 | 2.2% | 77.8% | Interannually and regionally varying seasonal patterns |

## 4.2 PC2: Damping

Whereas PC1 visualizes the differences between stream discharge and groundwater head time series (Fig. 2), PC2 provides information about the most important underlying process, explaining 11.3% of the variance of the total data set (Table 1).

Correspondingly, except for the sign, similar features in terms of distribution of loadings (Fig. 2) are found for PC1 and PC2. This applies to the different ranges of loadings for the two groups as well as for the overlapping ranges of loadings, indicating a smooth transition between these two groups rather than a clear distinction.

The effects of all other single principal components add to that of PC1. This is accounted for by comparing two synthetic time series, on the one hand, adding the effect of P2 to that of PC1, and on the other hand subtracting that of PC2 from PC1

(Fig. 3). As the share of variance is the square of the correlation coefficient, the chosen factors of $\pm 0.7 \approx \pm\sqrt{0.5}$ assign equal weight to PC1 and PC2. This implies an overestimation of the effect of PC2 compared to the loadings for most of the time series, but it helps to better understand the related effect.





**Figure 3: Effect of PC2 on modifying temporal dynamics of discharge and groundwater head time series (selected period; upper panel), autocorrelation function (lower left panel), and frequency spectrum (lower right panel).**

The upper panel of Fig. 3 shows a section of the total study period as an example. Adding PC2 values to those of PC1 results in a much smoother time course, attenuated peaks and clearly delayed recovery afterwards compared to negative loadings on PC2. The lower left panel represents the respective autocorrelations functions for the total 43 years study period. For positive loadings on PC2 the autocorrelation function exhibits a much more delayed decay compared to positive loadings, confirming that PC2 grasps in the first place the degrees of smoothness of the related time series (Table 1). There is a tendency of more pronounced stream discharge damping in larger catchments, indicated by a positive, although weak correlation between





catchment area and PC2 loadings (r = 0.18). Large positive loadings, indicating pronounced damping, are only found at sites
with large depth to groundwater. The reverse inference, however, does not apply. Correspondingly, loadings on PC2 are not
significantly correlated with depth to groundwater.

The smoothing effect can be described as a low-pass filtering effect: For the negative loading on PC2 about 60% of the
variance is assigned to the high-frequency range right of the steep slope of the annual cycle (Fig. 3, lower right panel). In
contrast, for the positive loading on PC2 nearly 70% of the variance is due to the low-frequency part beyond the annual cycle
(Fig. 3, lower right panel). The smoother a time series the higher the probability that subsequent values increase or decrease
monotonically which inevitably results in trends. In fact Fig. 4 shows a close relationship between PC2 loadings and sign
and strength of trends both for time series of discharge (r = -0.55) and of groundwater head (r = -0.83). To allow direct
comparison between both groups and to account for different ranges of stream discharge at different gauges, trend strength is
given in terms of standard deviation for the entire 43 year study period.


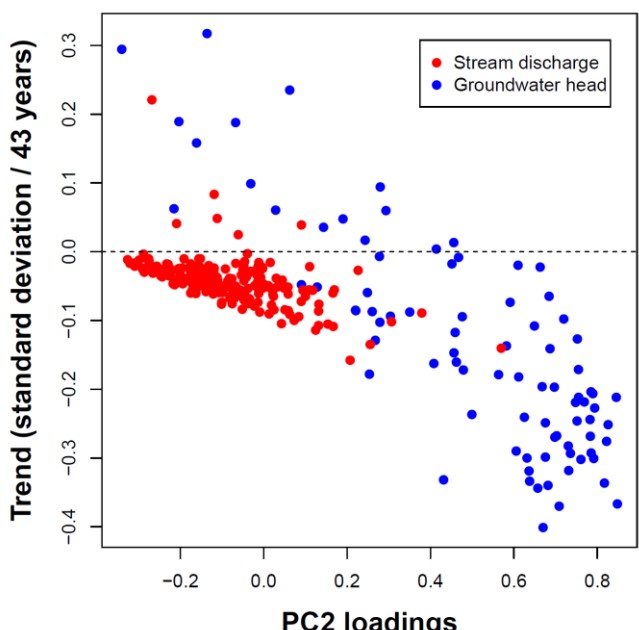

**Figure 4: Strength of trends 1980-2022 of stream discharge and groundwater head time series versus loading on PC2.**

### 4.3 PC3: West-east climate gradient

The third principal component covers 5.3% of the data set's variance (Table 1). In contrast to PC2, PC3 primarily depicts
differences between various stream discharge time series, whereas loadings are close to zero for all groundwater wells (Fig.
2). One important feature is the fact that depending on size and sign of PC3 loadings single peaks deflect either in the



positive or negative direction (Fig. 5, upper panel). In general, PC3 describes primarily short-living deviations from mean behaviour. Compared to other principal components (e.g., Fig. 3), the time series of PC scores exhibit very short

autocorrelation and reach the zero line after 3 weeks (not shown). In terms of seasonal patterns, positive loadings come along with a delayed decrease in spring (Fig. 5, lower left panel). Among all principal components PC3 loadings of streams exhibit the closest relationships with arable ($r = -0.73$) and forest ($r = 0.53$) land use in the respective catchments. More closely, though, correlate PC3 loadings with altitude ($r = 0.82$), annual mean precipitation ($r = 0.85$), number of snow days ($r = 0.90$), and annual mean potential evapotranspiration ($r = -0.84$). All of the aforementioned features are closely related with eastings

which in turn correlates with PC3 loadings ($r = 0.75$) (Fig. 5, lower right panel).



## Time series



**Figure 5: Effect of PC3 on modifying temporal dynamics of discharge and groundwater head time series (upper panel), seasonality (lower left panel), and spatial pattern of PC3 loadings (lower right).**


Due to pronounced collinearity between these different predictors multivariate linear regression was performed with stepwise elimination of non-significant predictors (p > 0.1). It was performed on the total data set as well as on the stream gauge and groundwater well data separately. Out of nine candidate predictors only longitude proved to be significant in all cases (Table 2). In addition, annual mean precipitation and annual mean potential evapotranspiration were identified as

significant predictors for PC3 loadings for the stream data, or for merged stream and well data. In contrast, none of the three





land use classes added predictive power to the model. In total, the regression model explained 25.5% of the variance of groundwater well loadings on PC3, and 31.6% for the other two models, respectively. It is concluded that PC3 reflects slight shifts of the temporal dynamics of meteorological observables from west to east, that is, roughly along the prevailing wind direction in this region (Table 1).


**Table 2: P values of various predictors for loadings on PC3 in multivariate regression models. "n.s." = not significant.**

|  | Steams and wells | Streams only | Wells only |
|---|---|---|---|
| Intercept | <0.001 | <0.001 | < 0.001 |
| Longitude | < 0.001 | < 0.001 | < 0.001 |
| Latitude | n.s. | n.s. | n.s. |
| Annual mean precipitation | 0.040 | 0.040 | n.s. |
| Annual mean potential evapotranspiration | 0.025 | 0.025 | n.s. |
| Number of snow days | n.s. | n.s. | n.s. |
| Percentage of arable fields | n.s. | n.s. | n.s. |
| Percentage of grassland | n.s. | n.s. | n.s. |

### 4.4 PC4: Porous versus hard rock aquifers

About 4.4% of the data set's variance is attributable to the fourth principal component (Table 1). Like for PC2, groundwater

head loadings exhibit a much wider span compared to streams (Fig. 2), indicating that the respective process affects groundwater more than streams. Negative loadings are associated with a very flashy behaviour, that is, pronounced single short-living spikes, but rapidly levelling off to an almost stable level within short time (Fig. 6, upper panel). In contrast, the time series with the positive loadings is characterized by a smoother behaviour with extended draining periods in the growing season, indicating a much more pronounced buffer capacity against falling dry (Fig. 6, upper panel).

Correspondingly, positive loadings come along with more pronounced memory effects, that is, higher autocorrelation, at the scale of up to 13 weeks. Zero crossing of the autocorrelation function after 3 and 9 months, a minimum after 6 months and a maximum at 12 months illustrate the pronounced seasonal pattern that is virtually absent for the negative PC4 loading which rapidly decays within two to three weeks (Fig. 6, lower left panel). This component seems to differentiate between two types of aquifers: Most porous aquifers exhibit positive loadings on PC4, and most fractured or karstic hard rock aquifers negative

loadings (Fig. 6, lower right panel). Loadings of unconsolidated and hard rocks differed significantly ($p < 0.05$; Wilcoxon test). Further differentiation of hard rock aquifers into karstic ($n = 15$) and fractured ($n = 30$) rocks did not reveal significant differences between these two groups. We conclude that this principal component provides strong evidence that karstic and



fractured rocks exhibit a lower water retention capacity compared to porous aquifers due to a lack of fine-grained material in the voids of consolidated rocks (Table 1).


**Figure 6: Effect of PC4 on modifying temporal dynamics of discharge and groundwater head time series (selected period; upper panel), autocorrelation function (lower left panel) and effect of aquifer type on PC4 loadings (lower right panel).**





### 4.5 PC5: Snow pack

The fifth principal component covers 3.4% of the variance of the data set (Table 1). Time series which load positively on this component are characterized by a delayed increase of discharge or groundwater head during the dormant season and a delayed decrease in spring and summer compared (Fig. 7, upper panel and lower left panel). In contrast, time series with strong negative loadings already reach the maximum plateau in January and drain rapidly after week 13 (Fig. 7, lower left panel).



**Figure 7: Effect of PC5 on modifying temporal dynamics of discharge and groundwater head time series (upper panel), mean seasonal patterns (lower left panel), and relationship between loadings and mean number of snow days per year (lower right panel).**

In terms of spatial patterns the mountainous region in the Southeast clearly stands out with high positive loadings (Fig. 7, lower right panel). Large parts of this region are at an altitude of more than 100 m a.s.l. (cf. Fig. 1) where perpetual snow packs develop for up to more than 180 days per year on average (Fig. 7, lower right panel). In contrast, a number of 100 snow days per year is exceeded only on single mountain tops along the northeastern and northern border of the study region, and is considerably less for the rest of the region. Thus a designation as a "snow pack component" for short is justified (Table 1).

### 4.6 PC6: Shift of seasonal patterns

The sixth principal component explains 2.2% of the variance of the data set (Table 1). It is slightly more relevant with respect to the spatial pattern of stream discharge compared to that of groundwater head time series. However, the difference is not as pronounced as for, e.g., PC2, PC3 or PC4 (Fig. 2).

The time series of PC6 scores exhibits two major features: On the one hand it exhibits single short spikes both in upward or downward direction, mostly during the dormant season (Fig. 8). On the other hand, it is characterized by a seasonal pattern with a maximum usually during the first half of the year, and a minimum in the second half (Fig. 8, upper panel). This results in a shift forward of the seasonal cycle for positive loadings, and a shift backward for negative loadings (Fig. 8, lower left panel). However, the exact timing of the seasonal pattern as well as the sign of component scores differs strongly between years. E.g., PC6 exhibits positive scores during the first four months of the year in 1990 (Fig. 8, upper panel), indicating a delay of early growing season depression of stream discharge and groundwater head (Fig. 8, lower left panel). In contrast, scores are negative in early 1992 (Fig. 8, upper panel), indicating an inverse effect. Later on PC6 scores are around zero for the whole growing season of 1992 (Fig. 8, upper panel), indicating no effect at all.



## Time series of PC6 scores



**Figure 8: Effect of PC6 on modifying temporal dynamics of discharge and groundwater head time series (upper panel), mean seasonal patterns (lower left panel), and relationship between loadings and mean number of snow days per year (lower right panel).**

In regards to spatial patterns, highest positive loadings were found in the northwestern part of the study region (Fig. 8, lower right panel). In addition, positive loadings prevail close to the western and eastern border. Here peak altitude exceeds 900 m a.s.l. in the Northwest (Rhön mountains) and 1000 m a.s.l. in the East (Fichtelgebirge mountains and Bavarian Forest mountains; Fig. 1). In contrast, negative loadings are abundant in the middle and southern part (cf. Fig. 1) where altitude is



only 656 m a.s.l. at maximum (Frankonian Alb). To summarise, PC6 seems to depict mainly shifts of the onset of the growing season, differing between years and regions (Table 1).

## 5 Discussion

We applied principal component analysis to a comprehensive data set of time series of groundwater head and stream discharge to identify the key drivers of spatial heterogeneity of hydrological behaviour in a 36,000 km$^2$ region. Whereas the determination of principal components is unambiguous, assigning meaning to the components in terms of processes and effects might always be a matter of debate. Note that the same approach can be used without any interpretation, e.g., to check for anthropogenic effects (Lehr and Lischeid, 2020). In contrast, in this study we did our best to constrain all interpretations

by using information on spatial patterns of loadings and impacts of components on temporal dynamics in the short term, in terms of seasonal patterns and of long-term behaviour. However, common definitions of processes do not necessarily translate directly to orthogonal components (Hannachi et al., 2007). On the other hand, it is very unlikely that orthogonal patterns as determined by PCA could develop in real-world data sets only by chance. In any case there is no hard proof for the presented interpretations. Thus all conclusions should be considered valid only until better explanations are found.

Only the first six principal components have been analysed in more detail, covering 77.8% of the total variance of the data set. The remaining components grasp only small shares of variance, that is, less than 2% each. In addition, these components tend to be less stable (Lehr and Lischeid, 2020) and often describe local patterns which are relevant for a small subset of sites only (Lischeid et al., 2021).

   About 51.1% of the variance of the observed groundwater head and discharge dynamics was described by one single

principal component which reflected the similarities of all time series. Correspondingly, the vast majority of observed time series exhibited positive correlation with this principal component (Fig. 2). Similar findings have been reported from other regions, e.g., by Lehr and Lischeid (2020) and Lischeid et al. (2021). Note that the approach followed in this study differs from the common application of Empirical Orthogonal Functions in climatology, where mean behaviour is subtracted from the observations and only the remaining residuals (called "anomalies") are subjected to the analysis.

Whereas the first component grasps the similarities, all other components describe deviations from that observed at single sites. Each of these components reflects specific modifications of common behaviour which is used for assignment to processes. Correspondingly, the distributions of loadings on these components are approximately symmetrical with respect to zero (Horel, 1981; Hannachi et al., 2007). In the following subsections the assignment of components to processes is discussed. Note that each site is characterized by a site-specific set of the contributions of the various principal components,

i.e., the loadings. The different effects captured by the principal components add onto each other. For example, the effects of climate patterns captured by PC3, PC5 and PC6 are further modified by PC2 and PC4, that is, the effects of signal propagation through the vadose zone. Thus adjacent sites will have similar loadings on the climate-related PC3, PC5 and PC6, but can differ considerably in regard to PC2 and PC4 due to small-scale variations of subsurface properties.





### 5.1 Climate patterns

Among the first six principal components PC3, PC5 and PC6 exhibited clearly larger ranges and higher variability of loadings of stream discharge time series compared to those of groundwater head (Fig. 2). This provides an indication that the respective processes affect stream discharge more than groundwater head which in turn points to processes located at or close to the input boundary of the hydrological system. We ascribed these components to various climate effects. All together they cover 10.9% of the total variance, or 21.2% of spatial variance, that is, from total variance after subtracting the

share of the first component.

Indices related to climate rank the highest among catchments indices used to understand the drivers of spatial variability of stream discharge dynamics (Tarasova et al., 2024; He et al., 2024; Addor et al., 2018) and to model stream discharge (Istalkar and Biswal, 2024). Correspondingly, based on remote sensing data, Joshi and Mohanty (2010) report that rainfall patterns explained 88% of the variance of observed topsoil moisture patterns in the subhumid Southern Great Plains region.

In contrast, empirical studies relating regional climate patterns to groundwater head dynamics are made difficult due to pronounced memory effects (Cuthbert et al., 2019) which mask subtle regional gradients. Thus, e.g., Longuevergne et al. (2007) followed a similar approach like in this study, that is, to firstly determine spatial patterns of groundwater head dynamics using principal component analysis (termed Karhunen-Loève transform in their study) and then relating these spatial patterns to those of climate and other effects.

In our study, roughly half of the variance covered by the climate components is accounted for by the third component (PC3). Whereas the map of loadings shows a clear west-east gradient, time series of positive compared to those of negative loadings do not exhibit clear systematic patterns. This lack of a clear pattern indicates that some major precipitation events are restricted either to the western or to the eastern part of the region (Fig. 5). Note that the PCA does not consider absolute values of discharge or groundwater head. Thus close correlation of PC3 loadings with annual mean values of precipitation,

potential evapotranspiration and number of snow days must not be interpreted as direct causal effects. Instead, it points indirectly to typical shifts in the temporal patterns related to the gradients of mean climatic values. One example is provided by Fig. 5 depicting a systematic delay of discharge and groundwater head peaks in spring and early summer in the eastern part compared to the western part which is ascribed to a slightly more nival runoff and groundwater recharge regime in the higher altitude regions in the East (Fig. 1).

Similar spatial shifts in the temporal dynamics of climatic variables affecting stream discharge and groundwater head dynamics have been found even in regions without corresponding topographical gradients in North Germany (Thomas et al. 2012, Lehr and Lischeid 2020, Lischeid et al. 2021). Two of these studies have been performed in overlapping regions: The PCA of stream discharge by Thomas et al. (2012) identified climatic gradients ranking first and second among the causes of spatial variability, and transformation of the hydrological input signals in the subsurface ranking third. In contrast, the latter

effect ranked first, and climatic gradients second and third in the study of groundwater head time series performed by



Lischeid et al. (2021). This confirms the statement at the beginning of this subsection that stream discharge is more sensitive to the dynamics of input signals, and that of groundwater head more to the subsurface transformation of the input signal.

The impacts of snow cover on hydrological dynamics are more clearly visible in PC5, i.e., in the more pronounced seasonal shift of runoff and groundwater recharge toward the summer and the spatial focus on high altitudes (Fig. 7) with a large
number of snow days. The latter presumably is only indirectly causally related to the seasonal shift described by PC5: Rather than the number of snow days per se the length of periods of perpetual snow pack is more relevant for the seasonal offset of groundwater recharge and stream discharge. Note that PC5 is obviously restricted to the effect of snow cover, whereas PC3 captures the effects of spatial gradients of other climatic variables like precipitation and potential evapotranspiration as well. Especially for high altitude sites in the eastern part of the study region both effects add to each other.

The last out of three climate-related components, PC6 (Fig. 8), does neither depict an annually recurring phenomenon like PC5 nor large scale spatial shifts of the dynamics of climate variables like PC3, but rather shifts of seasonal patterns that differ between sub-regions and between years. Thus it depicts some of the space-time variance of climatic variables in addition to that grasped by PC3 and PC5 but cannot directly be related to a single process.

## 5.2 Land use effects

Land use is commonly considered a major driver of spatial variance of stream discharge (Tarasova et al., 2024). It has often been shown and is widely agreed upon that built-up areas exert major impacts on stream flow (Mensah et al., 2022). In our study, however, the share of built-up areas did exceed 10% of catchment area only for 25 out of 292 sites. Correspondingly, we did not find any significant impact on principal component loadings. In the following we will focus on the effect of different vegetation types.

Our analysis considered only patterns in time but not absolute values. This limits the generalisability of our findings in terms of land use effects. On the other hand, though, we expected to find differing temporal patterns, e.g., an earlier onset of groundwater recharge and stream discharge in arable land after harvest of winter crops compared to forested areas. Although our analysis revealed a significant correlation of the share of arable land or forest cover with loadings on PC3, in particular for stream discharge, the effect of PC3 was not consistent with expected effects of land use. Rather than shifts in the seasonal
patterns that could be related to that of evapotranspiration of different land use classes PC3 only affected short-lived responses of flood peaks (Fig. 5). It would not be plausible that forests sometimes enhance, sometimes suppress flood peaks compared to arable fields. In addition various climate variables and altitude correlated much more closely with PC3 loadings compared to land use classes. Thus we conclude that the apparent effect of land use on stream discharge has to be ascribed to the dependence of land use on climate which in turn affects stream discharge rather than a direct causal relationship.

That does not imply that land use would not affect hydrological processes at all. Effects of crop type on topsoil moisture dynamics have been proven at the field scale (Zhao et al., 2010; Baroni et al., 2013; Hohenbrink et al., 2016; Scholz et al., 2024) and at the regional scale (Joshi and Mohanty, 2010), and on stream discharge at the regional scale (Thomas et al.,



2012). In contrast, Lehr and Lischeid (2020) and Lischeid et al. (2021) could not prove any effect on groundwater head dynamics at the regional scale.

The current literature reflects in fact contradictory findings in regard to land use effects on stream discharge. A plethora of modelling studies have been performed assuming clear effects (Dwarakish et al., 2015; Mensah et al., 2022). Thorough tests of that hypothesis, though, ranked land use effects fairly low compared to other effects (He et al., 2024; Duarte et al., 2024), or did not even find any clear effects at all (Istalkar and Biswal, 2024; Addor et al., 2018). Niehoff et al. (2002) found that land use affected runoff generation only under specific conditions. To conclude, and consistent with our findings, land use

effects on stream discharge seem to be rather weak and hard to identify in regional studies. This is presumably all the more true for land use effects on groundwater head dynamics.

### 5.3 Subsurface processes

Soil data are often considered decisive for hydrological processes (Tarasova et al., 2024; Vereecken et al., 2022) but have not been included here. This is partly due to the problems associated with the use of soil maps for hydrological studies

(Jakisch et al., 2021). In addition, detailed information of the complete vadose zone would have been needed, not only of the uppermost soil layers. Hohenbrink and Lischeid (2015) have shown that the effects of different texture classes in heterogeneous soils balance each other out, thus information of a single layer would be of little value for assessing groundwater recharge at greater depth. Gao et al. (2023) even go so far to state that "soil hydraulic properties are an effect rather than a cause of water movement", being optimised in long-term processes of co-evolution of soils and vegetation.

They see this as a major reason for the often found poor correlation between soil properties and hydrological processes. Our study considered only very limited information about subsurface properties, that is, a differentiation between thick unconsolidated sediments, fractured and karstified bedrock. Thus our data do not allow any inferences on the effects of soil properties. Neither does our analysis allow for any inferences on the role of surface runoff and interflow which might contribute to rapid stream response to heavy rainstorms.

Loadings of groundwater head time series on PC2 and PC4 exhibited much larger range and variability than those of stream discharge (Fig. 2). This points to processes at greater depth. But it does not necessarily imply that these processes would be irrelevant for a thorough understanding of the runoff dynamics. In total these two components explained 15.7% of the variance, that is, 31% of spatial variance when the contribution of PC1 is factored out. The second PC reflects the different degrees of damping of hydrological input signals, similar to the findings in other groundwater studies (Lehr and Lischeid,

2020; Lischeid et al., 2021). Mostly negative loadings of stream discharge time series on PC2 does not imply negative but much weaker damping compared to the average of all time series. This component is the only one, except PC1, that clearly separates stream discharge and groundwater time series (Fig. 2). However, Fig. 2 shows considerable overlap of loadings of these two groups, highlighting a continuum between discharge and groundwater head time series with smooth transitions in between. Although from a different perspective these results support the call by Berkowitz and Zehe (2020) to overcome the

separation of these "two water worlds".



Remarkably, damping of the hydrological signal does not increase continuously along the entire flow path from the vadose zone via groundwater to receiving streams. Rather, the strongest damping is observed only in groundwater at greater depth below surface which is usually at greater distance from the streams. Thus it can be concluded that damping is restricted to vertical seepage flux in the vadose zone but does not apply for lateral groundwater flow: Only the former exhibits

pronounced short-term dynamics whereas the latter is fairly stable at the scale of years and decades. The correlation between catchment area size and stream discharge loadings on PC2 might be due to a tendency to greater depth to groundwater being more prevalent in larger catchments than in small headwater catchments. But note that this correlation was found to be rather weak in this study.

The degree of damping observed in groundwater head time series has been found to be related to the thickness of the

unsaturated zone in other studies (Liesch and Wunsch, 2019; Lischeid et al., 2021). In this study, however, there was no significant correlation between depth to groundwater and PC2 loadings, in particular for shallow depth groundwater. The intensity of damping of hydrological signals in the subsurface depends on soil and vadose texture (Sawicz et al., 2011; Hohenbrink and Lischeid, 2015), soil structure, organic carbon content, etc. as well. In addition, any depth dependences are blurred in confined aquifers (Lischeid et al., 2017a). In view of these complexities it is remarkable that 11.3% of the variance

of temporal dynamics is captured by a single principal component.

However, the validity of a single principal component to grasp the damping of hydrological signals, as has been found by Hohenbrink and Lischeid (2015) and Lischeid et al. (2021) seems to be restricted to unconsolidated sediments. In this study a second component was identified that differentiated between unconsolidated sediments on the one hand and fractured or karstified bedrocks on the other hand (Fig. 6). The latter group does not only exhibit a lesser degree of damping, as

postulated, e.g., by Liesch and Wunsch (2019) and Olarinoye et al. (2022), but in particular exhibits sharp decreases in the falling limb of the hydrograph (Fig. 6), almost as a mirror image of the rising limb. Thus overall signal transformation resembles more that of water flow in a pipe rather than in a porous medium. Correspondingly the autocorrelation function rapidly falls off (Fig. 6). The same pattern has been found, e.g., for discharge in the highly karstic Baget catchment by Sivelle et al. (2022).

We did not find a significant difference between karstic limestone and fractured hardrock aquifers. This could partly at least be due to the small sample size and large within-groups variety. On the other hand there is no clear physical reason why water flow in large voids in karstic rocks should differ from those in fractured rocks. In any case this aspect deserves to be investigated in more detail in larger data sets.

## 5.4 Trends

We found diverging trends both for stream discharge and for groundwater head time series over the 43 years study period. These trends were closely related to PC2 loadings (Fig. 4), that is, the degree of smoothness of the time series. The same has been found for a region in North Germany by Lischeid et al. (2021) and Tsypin et al. (2025). These trends could neither be related to corresponding trends of climate variables nor to direct anthropogenic effects (Lischeid et al., 2021). Rather, the



smoother the time series the more became trends visible which were masked by short-term dynamics in less smooth time
series. Similar findings have been reported, e.g., for 215 wells in France by Baulon et al. (2022), and for nine wells with
more than 100 years data each in middle Europe by Liesch and Wunsch (2019). Correspondingly, Dong et al. (2019) found
more pronounced trends for groundwater level at greater depth in a data set comprising readings from more than 10,000
wells in the US. Boutt (2016) reported on significant trends only in 12% of precipitation stations, but in 60% of groundwater
wells in a region in New England (USA).

Pronounced trends found in particular in smooth time series visualize a more generic phenomenon that is masked by short-
term dynamics in less damped time series. In contradiction to common perceptions deep groundwater dynamics rather than
those of shallow groundwater or stream discharge should be regarded as an early warning in terms of climate change effects
on hydrological systems (Lischeid et al., 2021). That has to be accounted for when interpreting the results of trend analyses,
e.g., as is demanded by the European Water Directive. Whether a trend becomes clear highly depends on the presence or
absence of masking high-frequency patterns which are less abundant in deep groundwater head data.

Similar negative trends of deep groundwater head data have been observed in Northeast Germany (Lischeid et al., 2021) and
are consistent with trends of total water storage determined by the GRACE mission for Central and South Europe in the last
two decades (Xanke and Liesch, 2022; Kvas et al., 2024). These trends are traced back to climate drivers and are likely to
continue until 2100 at least (Wunsch et al., 2022). They seem to be driven more by an increase of evapotranspiration rather
than by a decrease of precipitation, as has been found by Sobaga et al. (2024) in 45 years lysimeter data in France, and by
Bruno and Duethmann (2024) in stream data all over Germany. These patterns being consistent at large scales confirm the
suitability of using deep groundwater head data as early warning tools in spite of the fact that the spatial density of
groundwater monitoring sites is often inferior to that of stream gauges, not the least due to construction costs and
accessibility.

Local effects like groundwater abstraction or land subsidence could modify that pattern at single sites. The data for this
study, however, have been provided from a groundwater monitoring network which had been designed explicitly to exclude
such effects.

## 6 Conclusions

A data set of weekly readings of stream discharge and groundwater from 292 sites, covering a 43 years period (1980-2022)
from a 36,000 km$^2$ region in South Germany was analysed to identify the prevailing drivers of spatial variance of
hydrological behaviour. The region is characterized by a large variety of land use and geological strata, comprising porous
aquifers as well as fractured and karstic hard rock aquifers. The first six components of a principal component analysis
explained 77.7% of the variance. Roughly half of the variance was covered by the first principal components which captured
the similarities of the set of time series.



According to the second principal component, differences between groundwater head and stream discharge were nearly exclusively due to different degrees of damping of the observed dynamics, with highest degrees observed at wells with high depth to groundwater table. The fourth component differentiated between porous aquifers on the one hand and fractured and karstic aquifers on the other hand, pointing to a more pipeflow-like behaviour in major conduits of the latter.

The higher the degree of damping, the more pronounced were negative trends both for groundwater head and stream
discharge for the whole 43 years period. These trends were attributed to a long-term increase of evapotranspiration. In contradiction to common perceptions the consequences were the most clearly discernible in smooth deep groundwater time series rather than in stream discharge were the long-term trends are masked by pronounced short-term responses to single events.

Three principal components (third, fifth and sixth) reflected various effects of climatic gradients on hydrological behaviour:
A general west-east gradient, which was driven by prevailing westerlies and a general topographic gradient; gradients of duration and height of snow cover, depending on altitude; and relative shifts of seasonality between years and sub-regions. These effects were more clearly visible in discharge than in groundwater head time series. Land use was closely related to topography and related to climatic patterns as well. Beyond this, though, no independent effect of land use on hydrological behaviour could be proven.

In spite of large heterogeneities in terms of geology, land use and related landscape features at the scale of the analysis the principal component analysis yielded surprisingly clear results. Groundwater head and stream discharge proved to be rather two poles along a common gradient of damping of the hydrological input signal rather than distinct systems. Spatial variance of stream discharge compared to that of groundwater head provided more information about the hydrological input dynamics due to climate effects. In contrast, spatial variance of groundwater head data was more closely related to subsurface
processes. Nevertheless, groundwater head dynamics rather than stream discharge revealed the consequences of an increase of evapotranspiration during the last decades. Thus a more pronounced understanding can be gained by merging stream and groundwater data which highlight different facets of the same hydrological system. We strongly recommend joint analyses to make more efficient use of existing monitoring data.

**Data availability**

Time series of stream discharge and groundwater head can be accessed via the state authority's web page at www.gkd.bayern.de. Specific requests, e.g., in terms of stream network data, should be addressed to datenstelle@lfu.bayern.de. Meteorological data can be downloaded from Deutscher Wetterdienst (2023; 2024). Digital elevation models of the study region are provided by Bundesamt für Kartographie und Geodäsie (2023; 2024).



**Author contribution**

GL conceptualized the study, developed the methodology, performed part of the formal analysis and visualization, and administrated and supervised the project. JW performed most of the formal analysis, of the programming, and part of the visualization. HS provided the data and critically reflected the results. GL prepared the manuscript with contributions from all co-authors.

**Competing interests**

The authors declare that they have no conflict of interest.

**Acknowledgement**

Provision of comprehensive high-quality data as well as the general support, collaboration and constructive feedback by various members of staff of the Bavarian State Authority of the Environment is highly appreciated.

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
