# Peer review of "What drives spatial variance of hydrological behaviour? An analysis of the regional groundwater-stream continuum"

_EGUsphere, 2025_

## Referee Comment (RC1)

Similar to clustering studies that meticulously compare cluster patterns, the paper is at times hard to follow for the reader, due to the tedious nature of going through detailed dissemination of specific patterns found in data. While this reflects the subjective impression of me as a reader, from the standpoint of the reviewer I can attest that the paper has a clear structure and exhibits straight and deep thinking about governing processes that is valuable for publication.

Its scientific significance is good: There are numerous studies tracing streamflow and groundwater time series patterns to overall governing processes – but few compare streamflow and groundwater side-by-side.

The scientific quality is fair, but can be good once the main general comment below addressing selective reporting of results is addressed.

Presentation quality is good and could be excellent if language and reasoning would be more concise and on point, and more aware of clearly communicating complicated mental concepts and connections instead of assuming the reader being "in the know" – at least in some parts of the results section. However, I would not draw necessity for improvement. It is good enough.

**General comments**

There is a fundamental problem of selective use of analysis methods in this paper: for example, the author does not report correlations of PC2-6 with raw time series means, but only the correlation of PC1 raw time series means is reported (line 173). As another example goes the analysis of autocorrelation (only PC2 & PC4, Figure 3 & Figure 6) or correlation with trends (only PC2, Figure 4) or the reporting of distribution of loadings w.r.t. aquifer type (only PC4, figure 6) or overall the selective reporting of correlations to individual catchment attributes throughout the paper, and several other examples. This is to say that in this paper, the (shown) application of methods is selective, and while selective reporting makes for a good storytelling, the conclusions draw in the study (PC1 captures "mean behavior", PC2 the "dampening" etc.) can only be drawn from comparative analysis, e.g. when correlation with time series mean is much better for PC1 than for PC2-6, and analogously for the other examples. It may or may not be that comparative analysis has been done in the background and only the significant results are shown (line 230 indicates that - stating that autocorrelation "not shown" for PC3). But to make sure that the results are not selective constructions, uniform method application and reporting across all PCs and is required. For the most part, at best in the appendix, to not blow up the paper, although some referencing in the main text of the manuscript will be needed. To be clear, I am not suggesting scientific misconduct. The displayed results seem to be reasonable, however for reasons of scientific rigor and for the reader to be able to reconstruct the conclusions properly, the paper simply needs verification via negative elimination through comparative analysis for reasons of completeness.

**Minor comments**

**Introduction section**

Line 33: WFD citation missing; maybe add half a sentence of explanation what WFD is for intercontinental audience; explain that for WFD the definition of water bodies includes GW

Line 33-45: In addition to citations to studies diagnosing the pitfalls of heterogeneities in groundwater, the author is advised to include citations to papers that also try to grasp it, e.g. this Gothenburg-based research group – but happily also others:

- Barthel, R., Haaf, E., Giese, M., Nygren, M., Heudorfer, B. and Stahl, K., 2021. Similarity-based approaches in hydrogeology: proposal of a new concept for data-scarce groundwater resource characterization and prediction. *Hydrogeology Journal*, 29(5), pp.1693-1709. https://doi.org/10.1007/s10040-021-02358-4
- Giese, M., Haaf, E., Heudorfer, B. and Barthel, R., 2020. Comparative hydrogeology-reference analysis of groundwater dynamics from neighbouring observation wells. Hydrological Sciences Journal, 65(10), pp.1685-1706. https://doi.org/10.1080/02626667.2020.1762888
- Haaf, E., Giese, M., Heudorfer, B., Stahl, K., & Barthel, R. (2020). Physiographic and climatic controls on regional groundwater dynamics. *Water Resources Research*, 56, e2019WR026545. https://doi.org/10.1029/2019WR026545

Line 79-80: for signature-based analyses of hydrologic similarity, essential (and more suitable) citations that need to be included to connect the reader to the large body of literature of earlier decades on hydrologic similarity, are the following; they are also good evidence for the bold claim that signatures are "hardly ever checked for relevance".

- Olden, J. D., & Poff, N. L. (2003). Redundancy and the choice of hydrologic indices for characterizing streamflow regimes. *River Research and Applications*, 19(2), 101–121. <a href="https://doi.org/10.1002/rra.700">https://doi.org/10.1002/rra.700</a>
- Olden, J.D., Kennard, M.J. and Pusey, B.J., 2012. A framework for hydrologic classification with a review of methodologies and applications in ecohydrology. *Ecohydrology*, 5(4), pp.503-518. https://doi.org/10.1002/eco.251

Line 68-71: This attempt can brilliantly be motivated based on literature as well, e.g. by this older call to interdisciplinary studies on hydro(-geo)logical similarity:

Barthel, R., 2014. HESS Opinions" Integration of groundwater and surface water research: an interdisciplinary problem?". *Hydrology and Earth System Sciences*, 18(7), pp.2615-2628. <a href="https://doi.org/10.5194/hess-18-2615-2014">https://doi.org/10.5194/hess-18-2615-2014</a>

Line 78-83: Maybe reiterate the method of choice (PCA) here.

**Data section**

Line 101-114: Would be informative to state the total number of samples and total number or share of interpolated samples that you had in there in the end.

Line 108: which analogous approach? Standard? Or citation? Unknown to me.

**Method section**

Line 130: "it has rarely been used in hydrology" -> that is simply untrue, as per the very large body of literature that opens up to the reader once they trace the literature starting from the Olden 2003/2012 paper provided above.

Line 140-143: Belongs to data section.

Line 145: "unlike in some other studies": cannot be stated like this without citations. Citation or rephrasing necessary.

**Results section**

See general comment about selective vs. comparative analysis

**Discussion section**

Reading through this discussion, I can only double my suggestion above to read the works of the referenced Gothenburg group, especially since they use data from the adjacent region. Haaf 2020 can be brilliantly connected to the overall discussion points, and the Giese 2021 paper to the subsurface section 5.3 especially, which is lacking discussion with literature overall. Also the Barthel 2014 paper belongs next to the Berkowitz and Zehe 2020 in line 444. Crazy how people who write about the same can be completely unaware of each other (Berkowitz/Zehe and Barthel, that is).

---

## Referee Comment (RC2)

**To Author:**

This research investigates the long-term drivers of streamflow and groundwater using principal components analysis. Overall, the idea of comparing hydrologic and hydrogeologic time series is an interesting study and more research should investigate the intersection of these similar disciplines.

The scientific rigor of the research is fair, but I'm not convinced that the results themselves lend significant insight into the drivers of streamflow and groundwater head. The PC6 explains only 2.2% of the variance and I'm not convinced this is significant enough to scientifically render a conclusion that PC6 is a driver of streamflow and groundwater. I think the PC6 is likely a contributor to a more complex system, but not a "driver" of the system.

It is not clear why p-values are displayed for some PCs, but not others. The statistical significance is needed to justify and support the results. Similarly, it is not clear why autocorrelation is presented for some but not all PCs. Finally, it is not clear why the trend is present for some but not all of the PCs.

All minor and major edits are described below:

**Introduction:**

- 1. The justification for the research itself is not clear. You describe in paragraph 1 and 2 that scientists are "tasked with relating heterogeneities", however, you do not explain the perceived heterogeneities or what would cause them. For example, since climate factors are included in your analysis and discussed later in the introduction, it would be relevant here to explain climate change as an important component of shifting hydrologic regimes. Additionally, you mention later that land use is included in your model and previous research therefore, it would be relevant to highlight the impact of land use on hydrology. Therefore, I am recommending either an extension of paragraph 1 or an additional paragraph where you discuss the overall cause of hydrologic heterogeneities (e.g. climate change, land use, both, others?). Additionally, provide an explanation of the relevance and importance of the research e.g. for watershed managers, water sustainability, etc.
- 2. Your introduction is primary focused on streamflow trends. I am recommending an extension of paragraph three (e.g. line 52) on the interconnectedness of streamflow and groundwater and the importance between the relationship to watershed hydrology. For example, you say in your hypothesis "Here we go a step further and hypothesize, that stream discharge and groundwater head dynamics are nothing than two poles along a common gradient, being subject to the same processes, although at different degrees." however, you need to provide an explanation on the background between the interconnectedness and for why this is important to study (also, including the literature). How are streamflow and groundwater head "two poles along a common gradient"?
- 3. The introduction does not explain or justify the importance of the study area for studying the streamflow/groundwater dichontomy. I am requesting an additional paragraph after line 63 discussing the relevance and importance of the study area.

4. Additionally, after the paragraph above, explain the previous literature on using PCA for this research and why it is the best statistical model to answer the question you pose. Why this statistical model and not another factoring model? Why is factoring the most appropriate model for this research?

**Data**

- 1. Line 85 100 should be the study area section
- 2. Line 100 should start the data section
- 3. In line 103 explain how the measurement sites were selected to reduce anthropogenic bias. You mention it but don't explain it provide a few more sentences explaining your process?
- 4. In line 103 what is the temporal resolution you say 43 full days was it daily, weekly, monthly, etc?
- 5. Line 118 Land use should be its own paragraph. Additional explanation is needed regarding what land use classifications are present, land use classifications used in the analysis, and a justification of the land use. Additionally, explanation of how the land use data were generated and whether they are raster or discrete data.
- 6. Line 120 more information needed on the raster climate data. 1. An explanation of how the data are created (from in situ data or modelled data?). What is the temporal resolution of the data?
- 7. With number of snow days what is the temporal resolution?
- 8. For potential evapotranspiration what is the unit?
- 9. Line 123 should probably go with the land use paragraph.
- 10. Line 152: An explanation of the parameters used in the PCA package did you do any manual parameterization or any model defined parameters?
- 11. Line 155 it is not clear what package was used for the PCA analysis. The packages provided here cover trends analysis, mapping, and plotting.

**Results:**

- 1. The principal components 1-6 explain 77.8% of the variance in the streamflow and groundwater head. It is interesting that each PC was explainable. However, I'm not convinced that a 2.2% variance is significant enough to justify a "driver". I think you need to 1. Provide literature and an iron-clad justification either in the methods or in the results that states your stance for including such low values 2. You need to call the PC with low explained variance something else e.g. not "drivers" but "contributors" and explain the new organization. These PC categories likely contribute to a complex system, but do not, among themselves act as "drivers"
- 2. Anywhere that the text mentions a "large correlation" a p-value and R/R2 should be included. For example, line 209 and line 211. Also the R/R2 values need to be svisualized in a table or figure and referenced in text.
- 3. Line 214: The r is shown and referred to figure 4 but the trend line and R value should be present on the figure.
- 4. It is not clear why p-values are displayed for some PCs, but not others. The statistical significance is needed to justify and support the results. Similarly, it is not clear why autocorrelation is present for some but not all PCs. Finally, it is not clear why the trend is present for some but not all of the PCs. Therefore, I am recommending that all the PCs

have a P-value table of loadings, an autocorrelation plot, and a trend plot (with R/R2 values).

**Discussion:**

1. I found the discussion to be quite good in describing the results in detail. However, if the introduction, methods, and results aren't updated as recommended above, your readers may never get to this section. The literature does a great job of pulling all the results together, however, most of the story and literature is not present in the introduction, therefore, making a clear disjunct/gap between the readers background knowledge, the understanding of the results, and the connection to the literature. Furthermore, the results themselves, as presented currently, do not support some of the literature and claims you make in the discussion. More statistical support is needed in the results for the readers to be truly convinced of the results that you're presenting.

---

## Author Comment (AC1)

**Review 1:**

Similar to clustering studies that meticulously compare cluster patterns, the paper is at times hard to follow for the reader, due to the tedious nature of going through detailed dissemination of specific patterns found in data. While this reflects the subjective impression of me as a reader, from the standpoint of the reviewer I can attest that the paper has a clear structure and exhibits straight and deep thinking about governing processes that is valuable for publication.

Its scientific significance is good: There are numerous studies tracing streamflow and groundwater time series patterns to overall governing processes – but few compare streamflow and groundwater side-by-side.

The scientific quality is fair, but can be good once the main general comment below addressing selective reporting of results is addressed.

Presentation quality is good and could be excellent if language and reasoning would be more concise and on point, and more aware of clearly communicating complicated mental concepts and connections instead of assuming the reader being "in the know" – at least in some parts of the results section. However, I would not draw necessity for improvement. It is good enough.

  ➢ Thanks for the thorough review of the paper and providing many helpful comments. We did our best to take up the criticisms to improve the paper.

**General comments**

There is a fundamental problem of selective use of analysis methods in this paper: for example, the author does not report correlations of PC2-6 with raw time series means, but only the correlation of PC1 raw time series means is reported (line 173). As another example goes the analysis of autocorrelation (only PC2 & PC4, Figure 3 & Figure 6) or correlation with trends (only PC2, Figure 4) or the reporting of distribution of loadings w.r.t. aquifer type (only PC4, figure 6) or overall the selective reporting of correlations to individual catchment attributes throughout the paper, and several other examples. This is to say that in this paper, the (shown) application of methods is selective, and while selective reporting makes for a good storytelling, the conclusions draw in the study (PC1 captures "mean behavior", PC2 the "dampening" etc.) can only be drawn from comparative analysis, e.g. when correlation with time series mean is much better for PC1 than for PC2-6, and analogously for the other examples. It may or may not be that comparative analysis has been done in the background and only the significant results are shown (line 230 indicates that – stating that autocorrelation "not shown" for PC3). But to make sure that the results are not selective constructions, uniform method application and reporting across all PCs and is required. For the most part, at best in the appendix, to not blow up the paper, although some referencing in the main text of the manuscript will be needed. To be clear, I am not suggesting scientific misconduct. The displayed results seem to be reasonable, however for reasons of scientific rigor and for the reader to be able to reconstruct the conclusions properly, the paper simply needs verification via negative elimination through comparative analysis for reasons of completeness.

  ➢ For each of the depicted principal components we provided only those pieces of information that we deemed helpful to test our hypotheses for plausibility, e.g., correlation with other variables, the effect of single principal components on autocorrelation or trends of the respective time series. These effects have been checked for all of the depicted principal components in the same way, but for the sake of brevity only the interesting results are presented in the manuscript. We will provide that information in the supplement.

**Minor comments**

**Introduction section**

Line 33: WFD citation missing; maybe add half a sentence of explanation what WFD is for intercontinental audience; explain that for WFD the definition of water bodies includes GW

➢ The reference will be included in the manuscript as well as an additional explanation: "The European Water Framework Directive (WFD) provides a legally binding framework for all member states of the European Union. It aims at achieving and maintaining a good water quality and good water quantity status in freshwater systems, coastal waters and groundwater bodies. Among others, the WFD demands a regular inspection of the "good quantitative" status of water bodies. To that end …"

Line 33-45: In addition to citations to studies diagnosing the pitfalls of heterogeneities in groundwater, the author is advised to include citations to papers that also try to grasp it, e.g. this Gothenburg-based research group – but happily also others:

- Barthel, R., Haaf, E., Giese, M., Nygren, M., Heudorfer, B. and Stahl, K., 2021. Similarity based approaches in hydrogeology: proposal of a new concept for data-scarce groundwater resource characterization and prediction. Hydrogeology Journal, 29(5), pp.1693-1709. https://doi.org/10.1007/s10040-021-02358-4

- Giese, M., Haaf, E., Heudorfer, B. and Barthel, R., 2020. Comparative hydrogeology– reference analysis of groundwater dynamics from neighbouring observation wells. Hydrological Sciences Journal, 65(10), pp.1685-1706. https://doi.org/10.1080/02626667.2020.1762888

- Haaf, E., Giese, M., Heudorfer, B., Stahl, K., & Barthel, R. (2020). Physiographic and climatic controls on regional groundwater dynamics. Water Resources Research, 56, e2019WR026545. https://doi.org/10.1029/2019WR026545

➢ We are sorry to have overlooked these papers (see more detailed comment to the Discussion section below) and will incorporate them in the manuscript.

Line 79-80: for signature-based analyses of hydrologic similarity, essential (and more suitable) citations that need to be included to connect the reader to the large body of literature of earlier decades on hydrologic similarity, are the following; they are also good evidence for the bold claim that signatures are "hardly ever checked for relevance".

- Olden, J. D., & Poff, N. L. (2003). Redundancy and the choice of hydrologic indices for characterizing streamflow regimes. River Research and Applications, 19(2), 101–121. https://doi.org/10.1002/rra.700

- Olden, J.D., Kennard, M.J. and Pusey, B.J., 2012. A framework for hydrologic classification with a review of methodologies and applications in ecohydrology. Ecohydrology, 5(4), pp.503-518. https://doi.org/10.1002/eco.251

➢ We are aware that there are some examples for a check for relevance but consider these examples rather scarce. We admit that we overlooked these papers (see more detailed

comment to the Discussion section below) and will incorporate them in the manuscript as counter examples.

Line 68-71: This attempt can brilliantly be motivated based on literature as well, e.g. by this older call to interdisciplinary studies on hydro(-geo)logical similarity:

- Barthel, R., 2014. HESS Opinions" Integration of groundwater and surface water research: an interdisciplinary problem?". Hydrology and Earth System Sciences, 18(7), pp.2615-2628. https://doi.org/10.5194/hess-18-2615-2014

➢ We are sorry to have overlooked this paper (see more detailed comment to the Discussion section below) and will incorporate it in the manuscript.

Line 78-83: Maybe reiterate the method of choice (PCA) here.

➢ Much of the relevant information including references to the literature is provided in the Method section. For the sake of brevity, we will add only two phrases here: "In numerous studies principal component analysis has proven its great potential to extract the prevailing features in large sets of interrelated variables or time series. This data-driven approach allows to differentiate between generic and site-specific features without any pre-defined assumptions."

**Data section**

Line 101-114: Would be informative to state the total number of samples and total number or share of interpolated samples that you had in there in the end.

➢ The number of dates per time series is given in line 109, the number of sites in line 113 – 114. Daily mean values of discharge have been calculated based on readings at 15 or 60 minutes intervals. In 44 out of 3,248,865 cases (15,695 days times 207 sites) one or more readings per day were missing. Only in one case did the gap cover a whole day, requiring interpolation between the preceding and the following day. Note that only 1/7 of these discharge data have been used eventually, that is, the values from every Monday. Groundwater data mostly exhibited daily resolution, except for seven wells with weekly intervals. However, data gaps were more abundant. In total, 75% of the groundwater measurement days matched exactly the final time axis of weekly data. All other had to be interpolated, usually over a time span of a few days. As described in l. 105- 108 the maximum length of the gap to be filled by interpolation was defined by the autocorrelation of the respective time series.

Line 108: which analogous approach? Standard? Or citation? Unknown to me.

➢ Standard autocorrelation analysis requires gapless time series with regular intervals and thus could not be applied to many of our time series. Thus we followed an approach analogously to that used in semivariogram analysis in geostatistics where irregular (spatial) spacing between data points is the rule rather than an exception: Pairwise correlation between time series is determined only for those lag widths where data were available, and then the results were summarized for different lag width classes.

**Method section**

Line 130: "it has rarely been used in hydrology" -> that is simply untrue, as per the very large body of literature that opens up to the reader once they trace the literature starting from the Olden 2003/2012 paper provided above.

> ➢ Principal Component Analysis is in fact widely used in hydrology to reduce the dimensionality of large sets of hydrograph or catchment indices. Olden et al. (2012) is a nice review in that regard. But that's not what this phrase is referring to. Rather, it refers to PCA application directly to a set of time series without the necessity of prior definition of any metrics. That approach is widely used, e.g., in climatology where it is coined as Empirical Orthogonal Function. But it is still not very common in hydrology. Otherwise we wouldn't encounter so many problems publishing respective papers.

Line 140-143: Belongs to data section.

> ➢ We consider data normalization an integral part of the parameterization of the PCA. PCA could be applied to non-normalized data as well, but that would change the interpretability of the results.

Line 145: "unlike in some other studies": cannot be stated like this without citations. Citation or rephrasing necessary.

> ➢ In fact weighting loadings by eigenvalues is the default setting of the R routine that we used. We wanted to make clear that our approach differed from that without going further into details which would go beyond the scope of the paper. We suggest to delete that clause.

**Results section**

See general comment about selective vs. comparative analysis

> ➢ See our reply to these comments.

**Discussion section**

Reading through this discussion, I can only double my suggestion above to read the works of the referenced Gothenburg group, especially since they use data from the adjacent region. Haaf 2020 can be brilliantly connected to the overall discussion points, and the Giese 2021 paper to the subsurface section 5.3 especially, which is lacking discussion with literature overall. Also the Barthel 2014 paper belongs next to the Berkowitz and Zehe 2020 in line 444. Crazy how people who write about the same can be completely unaware of each other (Berkowitz/Zehe and Barthel, that is).

> ➢ Although we do our best to keep an eye both on hydrological and hydrogeological research we have to admit that we overlooked these papers. Within the PUB framework catchment similarity has been studied and discussed extensively. But to the best of our knowledge it has been much less of a burning topic in hydrogeology so far. Thus, we focused our literature search on the hydrology community. Thanks for drawing our attention and that of future readers of the manuscript to this work. In spite of good will on either side bridging the gap

between these two sub-disciplines obviously is not without frictions when it comes to practice.

---

## Author Comment (AC2)

**Review 2:**

**To Author:**

This research investigates the long-term drivers of streamflow and groundwater using principal components analysis. Overall, the idea of comparing hydrologic and hydrogeologic time series is an interesting study and more research should investigate the intersection of these similar disciplines.

➢ Thanks for that appreciation, and for a thorough review of our paper.

The scientific rigor of the research is fair, but I'm not convinced that the results themselves lend significant insight into the drivers of streamflow and groundwater head. The PC6 explains only 2.2% of the variance and I'm not convinced this is significant enough to scientifically render a conclusion that PC6 is a driver of streamflow and groundwater. I think the PC6 is likely a contributor to a more complex system, but not a "driver" of the system.

➢ Our analysis aimed at extracting best-possible evidence for drivers of observed spatial patterns rather than at providing sound proofs. The latter would not have been possible anyhow at the spatial scale of the analysis irrespective of the share of variance. We assume that hydrological processes in heterogeneous landscapes are subjected to a large variety of various drivers. Thus we did not restrict our analyses to the most important two or three ones. But we did our best to underpin our hypotheses with detailed investigation of the spatial and temporal patterns of the respective principal components. Note that we use the term "driver" in regard to the observed patterns, not in terms of the driving physical forces of water transport.

It is not clear why p-values are displayed for some PCs, but not others. The statistical significance is needed to justify and support the results. Similarly, it is not clear why autocorrelation is presented for some but not all PCs. Finally, it is not clear why the trend is present for some but not all of the PCs.

➢ As stated below our analysis aimed at extracting best-possible evidence for drivers of observed spatial patterns rather than at providing sound proofs. For each of the depicted principal components we provided only those pieces of information that we deemed helpful to test our hypotheses for plausibility, e.g., correlation with other variables, the effect of single principal components on autocorrelation or trends of the respective time series. These effects have been checked for all of the depicted principal components in the same way, but for the sake of brevity only the interesting results are presented in the manuscript. We will provide that information in the supplement.

All minor and major edits are described below:

Introduction:

1. The justification for the research itself is not clear. You describe in paragraph 1 and 2 that scientists are "tasked with relating heterogeneities", however, you do not explain the perceived heterogeneities or what would cause them. For example, since climate factors are included in your analysis and discussed later in the introduction, it would be relevant here to explain climate change

as an important component of shifting hydrologic regimes. Additionally, you mention later that land use is included in your model and previous research – therefore, it would be relevant to highlight the impact of land use on hydrology. Therefore, I am recommending either an extension of paragraph 1 or an additional paragraph where you discuss the overall cause of hydrologic heterogeneities (e.g. climate change, land use, both, others?). Additionally, provide an explanation of the relevance and importance of the research – e.g. for watershed managers, water sustainability, etc.

> ➢ We regret that these aspects have not been clear enough and will modify the Introduction section accordingly. The motivation for this study is given in the first phrase of the Introduction section and is detailed in the following. The second phrase emphasizes the role of time series as the major source of information available to gain a better understanding of site-specific conditions and processes which are crucial for risk assessment for water resources management. Correspondingly, the term "heterogeneities" in line 29 refers to differences of temporal dynamics at different sites. Obviously, that was not clear enough and will be rephrased in the revised version of the manuscript. There is a huge body of literature on climate change effects on hydrology. We feel that scientists and water resources agencies are well aware of the key statements which do not need to be reflected here. Instead, the second paragraph focuses on the common approach to identify climate change (or other harmful) effects in given data sets and the problems this approach encounters in practice – which highlights the previous statement that we need to get a better understanding of the causes of different temporal dynamics at different sites. Paragraph 3 and 4 shortly reflect on the discussion in the literature on these possible causes. These include land use as well. Again we feel that readers of the journal are aware of the state of the art in this regard which does not need to be reflected here in more detail.

2. Your introduction is primary focused on streamflow trends. I am recommending an extension of paragraph three (e.g. line 52) on the interconnectedness of streamflow and groundwater and the importance between the relationship to watershed hydrology. For example, you say in your hypothesis "Here we go a step further and hypothesize, that stream discharge and groundwater head dynamics are nothing than two poles along a common gradient, being subject to the same processes, although at different degrees." – however, you need to provide an explanation on the background between the interconnectedness and for why this is important to study (also, including the literature). How are streamflow and groundwater head "two poles along a common gradient"?

> ➢ The second paragraph of the Introduction refers to trends both in stream discharge and groundwater head data. Note that line 37 and line 40-44 focus on groundwater explicitly. In contrast, the third paragraph focuses on streamflow only. But here the focus is not on trends but on numerous studies aiming at a better understanding of the different dynamics at different sites. Although this has been studied for groundwater head dynamics as well, much more work has been done for stream discharge recently. In regard to the interconnectedness of streamflow and groundwater we refer to the literature and will add more references in the revised manuscript. The term "two poles along a common gradient" refers to the temporal dynamics, be it flux rates or pressure heads. It assumes that there is no fundamental difference between these two types of time series other than different degrees of damping of the input signal. We will rephrase it in the revised manuscript.

3. The introduction does not explain or justify the importance of the study area for studying the streamflow/groundwater dichotomy. I am requesting an additional paragraph after line 63 discussing the relevance and importance of the study area.

➢ We will add a short paragraph: "Thus there is urgent need for studies in regions which exhibit not only a dense network of stream gauges and groundwater observation wells, but large gradients in regard to topography, climate, land use, and geology as well. This was the motivation for the selection of the study region (see below)."

4. Additionally, after the paragraph above, explain the previous literature on using PCA for this research and why it is the best statistical model to answer the question you pose. Why this statistical model and not another factoring model? Why is factoring the most appropriate model for this research?

➢ Much of this information including references to the literature is provided in the Method section. For the sake of brevity, we will add only two phrases here: "In numerous studies principal component analysis has proven its great potential to extract the prevailing features in large sets of interrelated variables or time series. This data-driven approach allows to differentiate between generic and site-specific features without any pre-defined assumptions."

Data

1. Line 85 – 100 should be the study area section

➢ We feel that these 16 lines do not justify a separate section. Thus we propose to change the heading to "Study region and data".

2. Line 100 should start the data section

➢ See comment above.

3. In line 103 – explain how the measurement sites were selected to reduce anthropogenic bias. You mention it but don't explain it – provide a few more sentences explaining your process?

➢ The sites have been selected by the authorities who provided the data based on their knowledge about local site conditions. The phrase will be re-formulated to make that clear.

4. In line 103 – what is the temporal resolution – you say 43 full days – was it daily, weekly, monthly, etc?

➢ "43" refers to the total length of the time series in years, not to the temporal resolution. The data were provided mostly as daily mean values, but with exceptions for some groundwater head data and for some periods. Daily mean values of discharge have been calculated based on readings at 15 or 60 minutes intervals. In 44 out of 3,248,865 cases (15,695 days times 207 sites) one or more readings per day were missing. Only in one case did the gap cover a whole day, requiring interpolation between the preceding and the following day. Note that only 1/7 of these discharge data have been used eventually, that is, the values from every

Monday. Groundwater data mostly exhibited daily resolution, except for seven wells with weekly intervals. However, data gaps were more abundant. In total, 75% of the groundwater measurement days matched exactly the final time axis of weekly data. All other had to be interpolated, usually over a time span of a few days. As described in l. 105- 108 the maximum length of the gap to be filled by interpolation was defined by the autocorrelation of the respective time series.

5. Line 118 – Land use should be its own paragraph. Additional explanation is needed regarding what land use classifications are present, land use classifications used in the analysis, and a justification of the land use. Additionally, explanation of how the land use data were generated and whether they are raster or discrete data.

> Land use data were taken from a common remote sensing product, the Corine rater data (l. 118-120). We will provide more details in the revised version of the manuscript, including spatial resolution and the merging of various land use classes. The justification of including land use data as drivers of spatial heterogeneity of temporal dynamics is given in the Introduction section (l. 55-57), including references to the literature. We will elaborate a little bit more on how these data were used to characterize groundwater wells and will shift l. 125 ("Correspondingly, land use was determined within a radius of 500 m around each groundwater well.") to this paragraph (see comment below).

6. Line 120 – more information needed on the raster climate data. 1. An explanation of how the data are created (from in situ data – or modelled data?). What is the temporal resolution of the data?

> The raster climate data were provided by the German Weather Service (DWD). Spatial resolution was 1 km. Data of the national network of weather stations operated by DWD were interpolated using altitude regression and Inverse Distance Weighting. More details are provided by the given reference (Deutscher Wetterdienst 2024).

7. With number of snow days – what is the temporal resolution?

> Temporal resolution is daily.

8. For potential evapotranspiration – what is the unit?

> The unit is mm per day. That information will be added in the manuscript.

9. Line 123 should probably go with the land use paragraph.

> We will elaborate a little bit more on how these data were used to characterize groundwater wells and will shift l. 125 ("Correspondingly, land use was determined within a radius of 500 m around each groundwater well.") to this paragraph (see comment above).

10. Line 152: An explanation of the parameters used in the PCA package – did you do any manual parameterization or any model defined parameters?

➢ No other parameter settings have been used than normalization to zero mean and unit variance as stated in line 140-142.

11. Line 155 – it is not clear what package was used for the PCA analysis. The packages provided here cover trends analysis, mapping, and plotting.

➢ For PCA the "prcomp" routine was used. That information will be added in the manuscript.

Results:

1. The principal components 1-6 explain 77.8% of the variance in the streamflow and groundwater head. It is interesting that each PC was explainable. However, I'm not convinced that a 2.2% variance is significant enough to justify a "driver". I think you need to 1. Provide literature and an iron-clad justification either in the methods or in the results that states your stance for including such low values 2. You need to call the PC with low explained variance something else – e.g. not "drivers" but "contributors" and explain the new organization. These PC categories likely contribute to a complex system, but do not, among themselves act as "drivers".

➢ Our analysis aimed at extracting best-possible evidence for drivers of observed spatial patterns rather than at providing sound proofs. The latter would not have been possible anyhow at the spatial scale of the analysis irrespective of the share of variance. We assume that hydrological processes in heterogeneous landscapes are subjected to a large variety of various drivers. Thus we did not restrict our analyses to the most important two or three ones. But we did our best to underpin our hypotheses with detailed investigation of the spatial and temporal patterns of the respective principal components. Note that we use the term "driver" in regard to the observed patterns, not in terms of the driving physical forces of water transport.

2. Anywhere that the text mentions a "large correlation" – a p-value and R/R2 should be included. For example, line 209 and line 211. Also the R/R2 values need to be visualized in a table or figure and referenced in text.

➢ We checked whether we always gave the r values when we reported on significant correlation between principal component loadings of all sites and landscape features but did not find any exception. Note that "loadings" of single observed time series on single principal components are in fact "correlations" as well but being limited to single sites which would not allow for a standard significance test. This is the case, e.g., in line 209. The term "correlation" in line 336 actually refers to the loading. As this might be misleading, we will replace it by "loading". In line 211 we report on the insignificant correlation between PC2 loadings and depth to groundwater, thus we refrained from giving the r value.

3. Line 214: The r is shown and referred to figure 4 – but the trend line and R value should be present on the figure.

➢ We guess you mean line 216-217 rather than l. 214 (Line 214 refers to the spectrum analysis in Fig. 3, lower right panel). We did not include the r values in Fig. 4 for the sake of clarity of the graph but provide that information in the text. Note that a modified Mann-Kendall test

was used for trend analysis that does not assume a linear but a monotonic trend. Thus any trend line would be misleading.

4. It is not clear why p-values are displayed for some PCs, but not others. The statistical significance is needed to justify and support the results. Similarly, it is not clear why autocorrelation is present for some but not all PCs. Finally, it is not clear why the trend is present for some but not all of the PCs. Therefore, I am recommending that all the PCs have a P-value table of loadings, an autocorrelation plot, and a trend plot (with R/R2 values).

➢ As stated above our analysis aimed at extracting best-possible evidence for drivers of observed spatial patterns rather than at providing sound proofs. For each of the depicted principal components we provided only those pieces of information that we deemed helpful to test our hypotheses for plausibility, e.g., correlation with other variables or the effect of single principal components on autocorrelation or trends of the respective time series. These effects have been checked for all of the depicted principal components in the same way, but for the sake of brevity only the interesting results are presented in the manuscript. We will provide that information in the supplement.

Discussion:

1. I found the discussion to be quite good in describing the results in detail. However, if the introduction, methods, and results aren't updated as recommended above, your readers may never get to this section. The literature does a great job of pulling all the results together, however, most of the story and literature is not present in the introduction, therefore, making a clear disjunct/gap between the readers background knowledge, the understanding of the results, and the connection to the literature. Furthermore, the results themselves, as presented currently, do not support some of the literature and claims you make in the discussion. More statistical support is needed in the results for the readers to be truly convinced of the results that you're presenting.

➢ We consider this a concluding comment that sums up the comments above. We hope that the modifications outlined above will substantially improve the comprehensibility of the paper.